# Adaptation of the Client Diagnostic Questionnaire for East Africa

Edith Kamaru Kwobah[1]*, Suzanne Goodrich[2], Jayne Lewis Kulzer[3], Michael Kanyesigye[4], Sarah Obatsa[5], Julius Cheruiyot[6], Lorna Kiprono[6], Colma Kibet[6], Felix Ochieng[5], Elizabeth A. Bukusi[5], Susan Ofner[7], Steven A. Brown[7], Constantin T. Yiannoutsos[8], Lukoye Atwoli[9,10], Kara Wools-Kaloustian[2]

1 Department of Mental Health, Moi Teaching and Referral Hospital, Eldoret, Kenya, 2 Division of Infectious Diseases, Indiana University School of Medicine, Indianapolis, Indiana, United States of America, 3 Department of Obstetrics, Gynecology, and Reproductive Sciences, University of California San Francisco, San Francisco, California, United States of America, 4 Mbarara University of Science and Technology, Mbarara, Uganda, 5 Centre for Microbiology Research, Kenya Medical Research Institute, Kisumu, Kenya, 6 Academic Model Providing Access to Care, Eldoret, Kenya, 7 Department of Biostatistics and Health Data Science, Indiana University School of Medicine, Indianapolis, Indiana, United States of America, 8 Department of Biostatistics and Health Data Science, Indiana University Fairbanks School of Public Health, Indianapolis, Indiana, United States of America, 9 Department of Mental Health and Behavioral Sciences, Moi University School of Medicine, Eldoret, Kenya, 10 Brain and Mind Institute and the Department of Internal Medicine, Medical College East Africa, Aga Khan University, Nairobi, Kenya

* eckamaru@gmail.com

**Data Availability Statement:** Data is available within our Supporting Documents.

## Abstract

Research increasingly involves cross-cultural work with non-English-speaking populations, necessitating translation and cultural validation of research tools. This paper describes the process of translating and criterion validation of the Client Diagnostic Questionnaire (CDQ) for use in a multisite study in Kenya and Uganda. The English CDQ was translated into Swahili, Dholuo (Kenya) and Runyankole/Rukiga (Uganda) by expert translators. The translated documents underwent face validation by a bilingual committee, who resolved unclear statements, agreed on final translations and reviewed back translations to English. A diagnostic interview by a mental health specialist was used for criterion validation, and Kappa statistics assessed the strength of agreement between non-specialist scores and mental health professionals' diagnoses. Achieving semantic equivalence between translations was a challenge. Validation analysis was done with 30 participants at each site (median age 32.3 years (IQR = (26.5, 36.3)); 58 (64.4%) female). The sensitivity was 86.7%, specificity 64.4%, positive predictive value 70.9% and negative predictive value 82.9%. Diagnostic accuracy by the non-specialist was 75.6%. Agreement was substantial for major depressive episode and positive alcohol (past 6 months) and alcohol abuse (past 30 days). Agreement was moderate for other depressive disorders, panic disorder and psychosis screen; fair for generalized anxiety, drug abuse (past 6 months) and Post Traumatic Stress Disorder (PTSD); and poor for drug abuse (past 30 days). Variability of agreement between sites was seen for drug use (past 6 months) and PTSD. Our study successfully adapted the CDQ for use among people living with HIV in East Africa. We established that trained non-specialists can use the CDQ to screen for common mental health and substance use disorders with reasonable accuracy. Its use has the potential to increase case identification, improve

**Funding:** This work was supported by the National Institutes of Health (#U01AI069911 to EKK, SG, JLK, MK, SO, JC, LK, CK, FO, EAB, SO, SAB, CTY, LA, KWK). The funders had no role in study design, data collection and analysis, decision to publish, or preparation of the manuscript.

**Competing interests:** The authors have declared that no competing interests exist.

linkage to mental healthcare, and improve outcomes. We recommend further studies to establish the psychometric properties of the translated tool.

## Introduction

Most currently available research instruments were developed in high income countries for English speakers [1]. Studies are increasingly involving more cross-cultural work with non-English-speaking populations, making the approach to translation and cultural validation of instruments a critical issue [2]. The goal of instrument translation and validation is to ensure that the constructs being measured are equivalent, both in terms of semantics and content, as well as the intent of the original instrument while also being culturally sensitive and appropriate [3]. Without a sound process of translation and validation there may be erroneous interpretation of the results [4].

There are several methods used for translation. One of the most common methods of instrument translation is the translation–back translation method, where the source document is translated into the target language by a bilingual person and then independently translated back into the original language [5]. Unfortunately, there are a limited number of publications describing the process of translation and validation utilized in various studies. Descriptions of this work would be useful in demonstrating the credibility of these instrument translations and the resultant research data.

In order to maintain rigor, in addition to simple translation, there is a need to validate translated instruments within the cultural context in which they are being used [2]. This is particularly true for instruments that assess dimensions of mental health, given that the expressions of mental health distress are likely very different between cultural contexts [6]. This can be particularly challenging when adapting instruments for use in countries that have a number of diverse languages and cultural groups, even when only one or two languages are used for most day-to-day interactions. In Kenya for example, there are more than 40 different languages spoken [7]. The country's official languages are English and Swahili, but most children learn the vernacular language (native language based on the tribe) as their first language. Swahili is frequently the second language learned by Kenyans, and hence becomes the most spoken language in the country. Similarly, in Uganda there are approximately 41 languages, with English being the official language of the country and Luganda being spoken widely, although not an official language [8].

The process of validation is important in order to ensure that the tool measures what it is intended to measure [9]. The main aspects of validation include face validity (the subjective assessment of the degree to which a measure appears to be related to a specific construct), content validity (the degree to which items in an instrument reflect the content universe to which the instrument will be generalized), construct validity (how well a concept is translated or transformed into a functioning and operating reality), and criterion-related validity (the extent to which a measure is related to an outcome) [10]. Criterion validity demonstrates how well one measure predicts an outcome for another measure, and in most cases the comparator is the gold standard in a given field [11]. For mental health and substance use diagnoses, an interview with a mental health professional is considered the gold standard [12].

The original version of the Client Diagnostic Questionnaire (CDQ) was developed in 2004 to assess for mental health and substance use disorders in HIV-infected adults in the United States who speak English [13]. The CDQ is designed to be administered by both clinicians and

lay-personnel with no mental health training in order to identify: mood disorders, anxiety, alcohol and drug use, posttraumatic stress disorder (PTSD), and thought disorder. The original validation study of the CDQ compared screening by lay counselors to independent mental health professionals. The comparison yielded a sensitivity, specificity, and overall accuracy of 91, 78, and 85% respectively, for establishing the presence of any psychiatric disorder in HIV-infected adults [13].

Since the development of the original CDQ, there has been limited literature published on its translation to local languages and validation within the different cultural contexts [14]. The objective of this paper therefore is to describe the process of translating and validating the CDQ for use in a multisite study in East Africa and to highlight the challenges encountered in these processes in order to provide lessons learned for other researchers and programs working in cross-cultural settings.

## Materials and methods

### Ethics statement

All participants gave a written consent. All methods were conducted in accordance with the Declaration of Helsinki. The study was approved by Moi University/MTRH Institutional Ethical Review Committee No IREC 201844], Kenya Medical Research Institute's Scientific Ethics Review Unit (SERU No. 3708) and Mbarara University of Science and Technology Research Ethics Committee (REC # 19/04-18) and the Uganda National Council of Science and Technology (HS 2450) in Uganda and Indiana University (IU IRB 1803601471).

### Project design

Instrument translation and validation was the first step in preparing for implementation of the Syndemics Study, a multisite cross-sectional study designed to explore how mental health and substance use shapes engagement and retention in care and the clinical outcomes of people living with HIV (PLHIV). Translation and the face, content and construct validation process took place between October 2017 and January 2018. The study was approved by Moi University/MTRH Institutional Ethical Review Committee (No IREC 201844), Kenya Medical Research Institute's Scientific Ethics Review Unit (SERU No. 3708) and Mbarara University of Science and Technology Research Ethics Committee (REC # 19/04-18) and the Uganda National Council of Science and Technology (HS 2450) in Uganda and Indiana University (IU IRB 1803601471).

### Setting

The study was conducted by the East African International Epidemiological Databases to Evaluate AIDS (EA-IeDEA) consortium. The EA-IeDEA consortium includes HIV care and treatment programs in Kenya, Tanzania and Uganda [15, 16]. For this study, the three participating sites were in Eldoret and Kisumu, Kenya and Mbarara, Uganda. Eldoret is home to many tribes with Kalenjin, Luhya and Kikuyu being the predominant languages spoken, but because of this cosmopolitan setting, the most commonly spoken language is Swahili [17]. The study took place at Moi Teaching and Referral Hospital (Academic Model Providing Access To Healthcare (AMPATH). The second Kenyan site in Kisumu, enrolled participants at Lumumba sub-county Hospital through the KEMRI-Center for International Health, Education and Biosecurity (formerly known as the Family AIDS Care & Education Services (FACES)). Patients in this area predominantly speak Dholuo. Participants in Uganda were enrolled through the Immune Suppression Syndrome Clinic at Mbarara Regional Referral

**Table 1. CDQ local language translation and face-validity determination at participating sites.**

| Location | Eldoret, Kenya | Kisumu, Kenya | Mbarara, Uganda |
|---|---|---|---|
| **Language of local CDQ adaptation** | Swahili[1] | Dholuo[2] | Runyankole -Rukiga[3] |
| **Group members conducting translations to local language and face validation[4]** | Psychiatrist (2) General Medicine clinician Psychologist Nurse Community representative | Psychiatrist Interpersonal psychotherapist General Medicine clinician (2) Mental Health study coordinator Translator | Psychiatrist Study PI Study coordinator |
| **Use of unique expert translator for local language back translation from to English.** | Yes | Yes | No[5] |
| **Psychiatrist evaluation of local language and back translated CDQ** | Yes (2) | Yes (1) | Yes (1) |

[1] S1 Questionnaire: Swahili CDQ Questionnaire

[2] S2 Questionnaire: Dholuo CDQ Questionnaire

[3] S3 Questionnaire: Runyankole-Rukiga Questionnaire

[4] All fluent in English and the language of local adaptation.

[5] The same translator was used

Hospital. Mbarara, in the Southwestern part of Uganda, is home to many tribes with Runyan-kole-Rukiga (a shared language of two southwestern Ugandan tribes) being the predominant language spoken.

## Translation and face validation

The original CDQ tool and the training manual are freely available online (chrome-extension://efaidnbmnnnibpcajpcglclefindmkaj/https://targethiv.org/sites/default/files/file-upload/resources/Client_Diagnostic_Questionnaire.pdf), hence we did not require a license or permission for its use, and there is no documented restriction to its modification. Each participating site assembled a multidisciplinary team which followed similar steps in preparing the CDQ for use at their site (Table 1). To obtain semantic equivalence to the original CDQ, an expert translator first translated the English CDQ into the local predominant regional language with emphasis on assuring that the sentence structure and word choice of the locally translated CDQ captured the same meaning as the original CDQ. The team, led by a psychiatrist, then reviewed the translated document to ensure not only semantic equivalence, but to assure the concepts of each question set were maintained from the original (conceptual equivalence). For example, the questions on depression were assessed and adapted to ensure that the words and examples represent the concept of depression in the local population. By utilizing a multidisci-plinary team, each member was able to bring his or her knowledge and experience discussing mental health issues with patients and to assure that the questions being asked were worded in such a way as to not offend or induce reluctance to answer. This was followed by a face valida-tion meeting by a bilingual committee where the team resolved conflicting statements and agreed on the final translation. The local language translations were then translated back into English. This back translated English version was checked by the team against the original ver-sion to ensure that no elements of the original English CDQ's meaning were lost.

## Criterion validation

The locally translated CDQ was administered by trained study Research Assistants (RAs) flu-ent in that language. The RAs were not previously trained in assessment of mental health and

substance use, except for the Mbarara RA who had experience as a mental health nurse. The RAs underwent study specific training on administration and scoring of the CDQ as per the original CDQ User's Guide [13].

The Syndemics study planned for enrollment of 600 participants (200 from each site). Based on prior work in culture adaptations of health-related questionnaires [18, 19] we selected a convenience sample of the first 30 participants from each site (90 in total) who agreed to participate in the study to determine the effectiveness of translation and to complete the criterion validation. Participants were read an informed consent form describing the study by RAs in their language of choice and then signed two forms, one for themselves and one for the study. The consent also specified that a participant may be selected to talk with a psychiatrist to answer additional questions about their substance use and mental health. RAs administered the appropriate locally translated CDQ to each participant. Standardized scoring as per the original English CDQ was used to determine a "Positive" score in 12 areas: major depressive syndrome; other depressive syndrome; panic syndrome; generalize anxiety syndrome; alcohol abuse in the past six months; alcohol abused in the past 30 days; drug abuse in the past six months; drug abuse in the past 30 days; post-traumatic stress disorder (PTSD); psychosis; receipt of professional mental health treatment or psychiatric medications in the past six months; or currently receiving professional mental health treatment or psychiatric medications. The first 30 participants at each site then met with a psychiatrist who was blinded to the results of the RAs administration of the locally translated CDQ. The psychiatrist conducted an interview guided by the DSM V criteria for diagnosis of the disorders captured in the CDQ. The psychiatrist provided their findings (positive or negative diagnosis) for each of the 12 domains assessed on the CDQ summary sheet, specifying if the patient fulfilled criteria for each diagnosis.

## Criterion validation analysis

For each disorder, we assessed the strength of agreement between the RA scores and the mental health professional's diagnosis by the percent which showed perfect agreement and by the Kappa statistic, which is a measure of agreement adjusted for random chance [20]. Landis and Koch suggested the following categories of strength of agreement for the Kappa statistic: < 0.00 Poor, 0.00–0.20 Slight, 0.21–0.40 Fair, 0.41–0.60 Moderate, 0.61–0.80 Substantial, 0.81–1.00 Almost Perfect [21]. Agreement was also assessed for the entire region and separately for each site. In instances where the psychiatrist's diagnosis and the RA's scores overwhelmingly aligned, Kappa was not estimable since it requires counts in all four cells of the cross tabulation of the RA's score with the psychiatrist's score.

The presence of any substance use and/or mental health disorder was defined as any positive diagnosis from amongst the 12 items. The psychiatrist's overall measure was modeled using a logistic model with explanatory variable of the Syndemic participant's overall measure. Sensitivity, specificity, positive predictive value (PPV) and negative predictive value (NPV) were obtained from the receiver operating curve (S1 and S2 Data).

## Results

The criterion validation analysis was undertaken on 90 participants, with 30 evaluated at each study site between February and August 2019. The median age of the participants was 32.3 years (IQR 26.5–36.3) and 58 (64.4%) were female. In the sample of 90 participants, the non-specialist diagnosis using the locally translated CDQ had an overall accuracy of 75.6% (Table 2). The sensitivity of the non-specialist locally translated CDQ diagnosis was 86.7%; specificity 64.4%; Positive Predictive Value (PPV) 70.9%; and Negative Predictive Value (NPV) 82.9%.

**Table 2. Correlation between non-specialist CDQ diagnosis and specialist diagnosis in local language.**

| Psychiatrist | Non-Specialist Locally Translated CDQ Administration Results | |
| --- | --- | --- |
| | Positive N (%) | Negative N (%) |
| Positive | 39 (43.3%) | 6 (6.7%) |
| Negative | 16 (17.8%) | 29 (32.2%) |

## Gray highlighted boxes represent concordance

For the 12 items which were scored by psychiatrists and non-specialists perfect agreement ranged between 79% and 98%. Using the categorical classification of Kappa mentioned by Landis and Koch [21], agreement was: substantial for major depressive syndrome, hazardous alcohol use during the past six months and alcohol abuse during the past 30 days; and moderate for other depressive disorders, panic disorder and psychosis. However, agreement was only fair for generalized anxiety, drug abuse in the past six months and PTSD; and poor for drug abuse during the past 30 days (Table 3).

We further investigated the level of agreement between the non-specialist CDQ diagnoses and the specialist diagnoses at each site (Table 4). In Eldoret the percent agreement between the CDQ results and the psychiatrist diagnostic evaluation ranged from 77% to 97%. The Kappa was not estimable for the questions on panic syndrome, and the utilization of mental health care or medication currently or in the past 6 months because there were only counts in two of the four cells in the cross tabulation. For major depressive syndrome, generalized anxiety disorder, alcohol abuse during the past 30 days and during the past 6 months and psychosis, Kappa showed moderate to almost perfect agreement. However, the Kappa was only fair for other depressive syndrome and PTSD and was slight or poor for drug abuse during the past 6 months and during the past 30 days.

In Kisumu, the percent agreement between the CDQ and psychiatrist was greater than 80% for all questions except generalized anxiety syndrome (67%]) and PTSD (70%). The Kappa was not estimable for questions on the utilization of mental health care or medication currently or in the past 6 months because there was complete agreement between the CDQ and the specialist diagnoses. Kappa was moderate to almost perfect for major depressive disorder, other depressive syndrome, panic syndrome, alcohol abuse during the past 6 months, alcohol abuse during the past 30 days, drug abuse during the past 6 months, and psychosis. The questions on generalized anxiety syndrome and PTSD had fair agreement, while drug use during the past 30 days had poor agreement.

In Mbarara the non- specialist CDQ scores and psychiatrist diagnoses had perfect percent agreement which ranged between 87% and 100% for all questions. Kappa was not estimable for major depressive syndrome, panic syndrome, drug abuse during the past six months, drug abuse during the past 30 days, PTSD, psychosis, and the utilization of mental health care or medication currently or in the past 6 months. For Drug use in the past 6 months and 30 days, psychosis, and client utilization of mental health services in the past 6 months there was complete agreement between the locally translated CDQ and the Psychiatrist. For major depressive syndrome [1], panic syndrome [1], and PTSD [4] the locally translated CDQ identified cases which the psychiatrist did not, while for the question on patients currently receiving services the psychiatrist identified one case while the CDQ did not. There was moderate or substantial agreement for other depressive syndromes, alcohol abuse in the past 6 months and the past 30 days while there was only fair agreement for generalized anxiety syndrome.

**Table 3. Correlation between non-specialist CDQ diagnosis and specialist diagnosis in local language.**

| Diagnosis | Psychiatrist Diagnosis | Non-Specialist CDQ Diagnosis | | Perfect Agreement | Kappa (Std Error) | 95% CI |
|---|---|---|---|---|---|---|
| | | No N [%] | Yes N [%] | | | |
| Major Depressive Syndrome | No | 79 (87.8) | 4 (4.44) | 95.6 | 0.75 (0.1) | 0.53, 0.98 |
| | Yes | 0 (0.0) | 7 (7.8) | | | |
| Other Depressive Syndrome | No | 60 (66.7) | 8 (8.9) | 82.2 | 0.52 (0.11) | 0.31, 0.72 |
| | Yes | 8 (8.9) | 14 (15.6) | | | |
| Panic Syndrome | No | 74 (82.2) | 4 (4.4) | 88.9 | 0.48 (0.14) | 0.21, 0.76 |
| | Yes | 6 (6.7) | 6 (6.7) | | | |
| Generalized Anxiety Syndrome | No | 64 (71.1) | 14 (15.6) | 80 | 0.36 (0.12) | 0.13, 0.59 |
| | Yes | 4 (4.4) | 8 (8.9) | | | |
| Alcohol Abuse, past 6 months | No | 69 (76.7) | 5 (5.6) | 93.3 | 0.79 (0.08) | 0.63, 0.95 |
| | Yes | 1 (1.1) | 15 (16.7) | | | |
| Alcohol Abuse, past 30 days | No | 78 (86.7) | 6 (6.7) | 93.3 | 0.63 (0.13) | 0.37, 0.90 |
| | Yes | 0 (0.0) | 6 (6.7) | | | |
| Drug Abuse, past 6 months | No | 80 (88.9) | 1 (1.1) | 91.1 | 0.30 (0.18) | -0.04, 0.64 |
| | Yes | 7 (7.8) | 2 (2.2) | | | |
| Drug Abuse, past 30 days | No | 85 (94.4) | 2 (2.2) | 94.4 | -0.03 (0.01) | -0.05, 0.00 |
| | Yes | 3 (3.3) | 0 (0.0) | | | |
| PTSD | No | 64 (71.1) | 16 (17.8) | 78.9 | 0.32 (0.11) | 0.10, 0.54 |
| | Yes | 3 (3.3) | 7 (7.8) | | | |
| Psychosis | No | 83 (92.2) | 2 (2.2) | 95.6 | 0.58 (0.19) | 0.20, 0.95 |
| | Yes | 2 (2.2) | 3 (3.3) | | | |
| Received professional mental health treatment or prescribed psychiatric medications in the past 6 months | No | 88 (97.8) | 0 (0.0) | 97.8 | - | - |
| | Yes | 2 (2.2) | 0 (0.00) | | | |
| Currently receiving mental health treatment or psychiatric medications | No | 87 (96.7) | 0 (0.0) | 96.7 | - | - |
| | Yes | 3 (3.3) | 0 (0.0) | | | |
| Any Disorder | No | 29 (32.2) | 16 (17.8) | 75.6 | 0.51 (0.09) | 0.34, 0.68 |
| | Yes | 6 (6.7) | 39 (43.3) | | | |

Kappa Statistic Strength of: Poor: < 0.00, Slight: 0–20%, Fair:21–40%

Moderate: 41–60%, Substantial: 61–80%, Almost Perfect: 81–100%.

**Table 4. Correlation between non-specialist CDQ and specialist diagnosis in local language by site.**

| Diagnosis | Eldoret, Kenya | | | Kisumu, Kenya | | | Mbarara, Uganda | | |
|---|---|---|---|---|---|---|---|---|---|
| | Perfect Agreement | Kappa (Std Error) | 95% Confidence Interval | Perfect Agreement | Kappa (Std Error) | 95% Confidence Interval | Perfect Agreement | Kappa (Std Error) | 95% Confidence Interval |
| Major Depressive Syndrome | 93.3 | 0.76 (0.16) | 0.45, 1.00 | 96.7 | 0.84 (0.1) | 0.53, 1.00 | 96.7 | - | - |
| Other Depressive Syndrome | 76.7 | 0.39 (0.19) | 0.02, 0.75 | 83.3 | 0.59 (0.16) | 0.28, 0.91 | 86.7 | 0.58 (0.19) | 0.22, 0.95 |
| Panic Syndrome | 90.0 | - | - | 80.0 | 0.52 (0.17) | 0.19, 0.86 | 96.7 | - | - |
| Generalized Anxiety Syndrome | 86.7 | 0.43 (0.23) | -0.03, 0.89 | 66.7 | 0.26 (0.18) | -0.08, 0.61 | 86.7 | 0.29 (0.23) | -0.16, 0.75 |
| Alcohol Abuse during the past 6 months | 93.3 | 0.85 (0.10) | 0.65, 1.00 | 90.0 | 0.71 (0.15) | 0.41, 1.00 | 96.7 | 0.65 (0.32) | 0.02, 1.00 |
| Alcohol Abuse during the past 30 days | 96.7 | 0.78 (0.21) | 0.37, 1.00 | 86.7 | 0.53 (0.19) | 0.16, 0.91 | 96.7 | 0.65 (0.32) | 0.02, 1.00 |
| Drug Abuse during the past 6 months | 76.7 | 0.13 (0.18) | -0.22, 0.49 | 96.7 | 0.65 (0.32) | 0.02, 1.00 | 100 | - | - |
| Drug Abuse during the past 30 days | 90.0 | -0.05 (0.03) | -0.11, 0.02 | 93.3 | -0.03 (0.02) | -0.08, 0.01 | 100 | - | - |
| PTSD | 80.0 | 0.38 (0.20) | -0.02, 0.78 | 70.0 | 0.31 (0.16) | 0.00, 0.62 | 86.7 | - | - |
| Psychosis | 93.3 | 0.47 (0.31) | -0.13, 1.00 | 93.3 | 0.63 (0.23) | 0.18, 1.00 | 100 | - | - |
| mental health treatment/ psych medications, in past 6 months | 93.3 | - | - | 100 | - | - | 100 | - | - |
| Current mental health treatment/ psych medications | 93.3 | - | - | 100 | - | - | 96.7 | - | - |
| Any Disorder | 73.3 | 0.43 (0.17) | 0.10, 0.76 | 76.7 | 0.44 (0.17) | 0.12, 0.77 | 76.7 | 0.47 (0.17) | 0.14, 0.80 |

Kappa Statistic Strength of Agreement Categories: Poor: < 0.00, Slight: 0.00–0.20

Fair: 0.21–0.40, Moderate: 0.41–0.60, Substantial: 0.61–0.80, Almost Perfect: 0.81–1.00.

## Discussion

In this paper we describe the process of translation of the CDQ from English into three local languages for use at sites in East Africa, as well as the criterion validity of the translated CDQ. Investing time and expertise in the translation and cultural adaptation of a sensitive and complex questionnaire is an essential component of the validation process, as it sets the foundation for more accurate collection and interpretation of data within the local context [22]. We used an approach which was previously found to be effective and efficient by a South African study to establish semantic and content equivalence. This approach included combining back translation and a bilingual committee to resolve disagreements of word meaning or cultural appropriateness [23]. Our approach to adaptation resulted in several lessons learned.

First translational ambiguity is likely to occur in which more than one translation is possible for a given word [24], and this needs to be acknowledged and addressed by the bilingual committee. In some instances, we chose to provide more than one word in the translated document to make understanding clear. For example, in translating "alcoholic drink" into Swahili, we used two words: "pombe, the traditional Swahili word and "vileo, a more commonly used word.

Second, semantic equivalence is often challenging, requiring translators to interrogate literal translations of some words to avoid misinterpretations. Other times word translation has

to be done pragmatically, rather than by searching for word equivalence [25]. For example, it was difficult to find a single word in Swahili or Luo that is equivalent to "depression," or words for drugs of abuse (such as cocaine or heroin). Therefore, these words were not directly translated from English and explanations of the concept were provided by the RA if the participant did not understand the word. Additionally, once translated, it was difficult to differentiate anxiety, nervousness and feeling frightened in Swahili. This necessitated the use of a phrase rather than a single word to create equivalency with the original word in English. A Kenyan linguist described these translational challenges and explored the difficulties experienced by translators and interpreters as they strove to find equivalences during translation and interpretation exercises. He concluded that equivalence at sentence level-semantic may not be enough and requires exploring several approaches in order to retain the original meaning [26]. Within the three languages and cultures studied within this project all 12 concepts represented within the CDQ could be explored after appropriate semantic and conceptual equivalence is assured. If the CDQ is used outside of the cultures and languages studied within this project it is possible that one or more of the concepts more significant adaptations would be required if the concept does not exist in any form within the culture or language [27].

Our validation study of the CDQ translated tool showed good overall agreement between the non-specialist and the psychiatrist's diagnosis. The non-specialist could accurately diagnose mental health disorders 76% of the time with perfect agreement between 79% and 98% for the majority the items. This was lower than the 85% accuracy seen in the use of the original English version, but is comparable to the South African study which reported an accuracy of 79% [28]. Our East African language versions of the CDQ had a sensitivity and specificity of 87% and 65% as compared to the original English version's 91% and 65% respectively. The high sensitivity reached with the locally translated CDQ is significant in that identifying those patients who do have a substance use or mental health disorder is the most important part of a screening process. While specificity is lower, an over diagnosis of disorders can be limited by referrals to mental health professionals for confirmation. Psychiatrists benefit from the advantage of having open conversations with patients and being able to ask additional probing questions that can lead to determining if a disorder is present. Those administering a screening instrument such as the CDQ are limited in the information that is gathered which is the likely reason for their decreased accuracy.

In the analysis of agreement by site, we also observed some differences between non-specialist diagnosis and specialist diagnosis. For example, in diagnosing drug use during the past six months in Eldoret, Kenya, the psychiatrist identified six instances of drug use in the past six months where the non-specialist did not. This may be attributed to underreporting on the part of the patient due to a lack of awareness of drug use as a major health problem, or with shame due to use [29]. A clinician who is aware of this known reporting bias [30], is more likely to probe further and get better information than a non-specialist. The difference in diagnosis may also be related to the psychiatrist identifying tobacco as a drug, where the non-specialist did not. Another example of variation in agreement is seen in the diagnosis of PTSD. Non-specialists found PTSD in 13% to 27% of patients using the CDQ, where the psychiatrists did not. Given that these interviews were conducted closely together, it is possible that a participant would withhold trauma related information that triggers negative emotions, or through additional questioning psychiatrists were able to exclude the diagnosis [31]. Other possible contributors to the differences between sites may be the difference in the languages used, cultural understanding of mental health issues in the different countries and cities, or the clinician's level of experience. Given these variations in diagnosis it is important that both non-specialists and specialists are aware of these factors influencing outcomes. Repetition of the CDQ or the psychiatrist interview may be considered later if there is doubt regarding the

validity of the outcome. This may help minimize the chance of missing clients who would otherwise have benefitted from care.

The findings of our study must be interpreted with some limitations in mind. First, this was an observational study with a relatively small sample size. Second, given that we include three different sites, there may have been inter-rater variability between the different sites, as we did not conduct an inter-rater reliability test. Lastly, each site had different a psychiatrist interviewing patients and acting as the "gold standard" in giving the participant their diagnoses. The psychiatrists did not use a structured clinical interview tool, but rather their clinical skills on assessing conditions described in the DSM V. Interviews such as these, where sensitive questions are being asked and uncomfortable conversations may take place, the rapport that psychiatrist establishes with the patient may vary widely. Patients may bring fear, distrust, stigma, or shame with them to the conversation and the psychiatrists' ability to navigate these issues to get truthful answers in order to make a diagnosis are likely different. Recognition in the psychiatric community of the need to minimize differences in reliability have been discussed in the literature for many years and have led to strategies to improve reliability [32, 33]. Use of defined diagnostics criteria, standard definitions of symptoms and use of structured interviews is promoted to minimize differences in psychiatric evaluations. These issues account for some variations in the sites but do reflect the actual practices used and the challenges encountered in making these diagnoses. Despite this weakness, the strength of this study was the rigorous translation process and the involvement of local language experts in creating culturally and linguistically sound translations of the CDQ. In addition, the study was conducted at three sites with distinct cultural contexts and languages allowing for more generalizability than a single site study. Given that there are few mental health specialists in this region, yet increasing mental health needs, this study supports the idea that assessment and diagnosis by lay workers may be a feasible approach to increase screening and linkage to appropriate mental health services [34].

## Conclusion

Our study gives a description of the of the adaptation process of the previously validated Client Diagnostic Questionnaire for use in East Africa. Our study adds to the evidence that trained non-specialists can use this tool to screen for common mental health and substance use disorders with reasonable accuracy, hence increase opportunities for case identification and linkage to care, and this has potential to reduction in treatment gap and hence better outcomes among patients living with HIV. We recommend further studies to establish the psychometric properties of the translated tool.

## Supporting information

**S1 Questionnaire. Locally adapted Client Diagnostic Questionnaire (CDQ)- Kenya/Swahili.**
(DOCX)

**S2 Questionnaire. Locally adapted Client Diagnostic Questionnaire (CDQ-Kenya/Luo).**
(DOC)

**S3 Questionnaire. Locally adapted Client Diagnostic Questionnaire (CDQ–Uganda/Runyankole-Rukiga).**
(PDF)

**S1 Data. Project data set in SAS.**
(DOCX)

**S2 Data. Project data set notes.**
(CSV)

## Acknowledgments

National Institutes of Health-National Institute of Allergy and Infectious Diseases (NIH-NIAID) for supporting the IeDEA consortium and Cosmas Apaka and Marion Achieng, the country coordinators of East African Regional IeDEA Consortium, for coordinating all the research Activities.

## Author Contributions

**Conceptualization:** Edith Kamaru Kwobah, Suzanne Goodrich, Jayne Lewis Kulzer, Michael Kanyesigye, Elizabeth A. Bukusi, Susan Ofner, Steven A. Brown, Lukoye Atwoli, Kara Wools-Kaloustian.

**Data curation:** Sarah Obatsa, Julius Cheruiyot, Lorna Kiprono, Colma Kibet, Felix Ochieng.

**Formal analysis:** Susan Ofner, Steven A. Brown, Kara Wools-Kaloustian.

**Funding acquisition:** Constantin T. Yiannoutsos, Kara Wools-Kaloustian.

**Investigation:** Edith Kamaru Kwobah, Suzanne Goodrich, Jayne Lewis Kulzer, Sarah Obatsa, Julius Cheruiyot, Lorna Kiprono, Colma Kibet, Felix Ochieng, Susan Ofner, Steven A. Brown, Constantin T. Yiannoutsos, Lukoye Atwoli, Kara Wools-Kaloustian.

**Methodology:** Edith Kamaru Kwobah, Suzanne Goodrich, Elizabeth A. Bukusi, Susan Ofner, Steven A. Brown, Constantin T. Yiannoutsos, Lukoye Atwoli, Kara Wools-Kaloustian.

**Project administration:** Edith Kamaru Kwobah, Constantin T. Yiannoutsos, Kara Wools-Kaloustian.

**Resources:** Kara Wools-Kaloustian.

**Supervision:** Suzanne Goodrich, Elizabeth A. Bukusi, Kara Wools-Kaloustian.

**Validation:** Edith Kamaru Kwobah, Suzanne Goodrich, Michael Kanyesigye, Lukoye Atwoli, Kara Wools-Kaloustian.

**Visualization:** Suzanne Goodrich, Kara Wools-Kaloustian.

**Writing – original draft:** Edith Kamaru Kwobah.

**Writing – review & editing:** Edith Kamaru Kwobah, Suzanne Goodrich, Jayne Lewis Kulzer, Michael Kanyesigye, Sarah Obatsa, Julius Cheruiyot, Lorna Kiprono, Colma Kibet, Felix Ochieng, Elizabeth A. Bukusi, Susan Ofner, Steven A. Brown, Constantin T. Yiannoutsos, Lukoye Atwoli, Kara Wools-Kaloustian.

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
