## [Decision Letter · Decision Letter 0]

30 Aug 2023

PGPH-D-23-00103

Adaptation of the Client Diagnostic Questionnaire for East Africa

Dear Dr. Kwobah,

Thank you for submitting your manuscript to PLOS Global Public Health. After careful consideration, we feel that it has merit but does not fully meet PLOS Global Public Health’s publication criteria as it currently stands. Therefore, we invite you to submit a revised version of the manuscript that addresses the points raised during the review process.

We look forward to receiving your revised manuscript.

Kind regards,

Abhijit Nadkarni

Academic Editor

Journal Requirements:

1. Our staff editors have determined that your manuscript is likely within the scope of our Global Mental Health: challenges, opportunities, and the future of the field. This editorial initiative is headed by a team of Guest Editors for PLOS GPH: Rochelle Burgess (University College of London) and Dixon Chibanda (University of Zimbabwe and London School of Tropical Medicine and Hygiene). The Collection invites researchers to submit original research which engages with, or disrupts, the urgent needs across the global mental health landscape. We especially encourage submissions of studies that critically interrogate the status quo of the field and that involve inter-/trans-disciplinary approaches and those which share perspectives from underrepresented global regions and communities.

Additional information can be found on our announcement page: https://collections.plos.org/call-for-papers/global-mental-health-opportunities-challenges/ 

If you would like your manuscript to be considered for this collection, please let us know in your cover letter and we will ensure that your paper is treated as if you were responding to this call.  Please note that being considered for the Collection does not require additional peer review beyond the journal’s standard process and will not delay the publication of your manuscript if it is accepted by PLOS GPH. If you would prefer to remove your manuscript from collection consideration, please specify this in the cover letter."

3. Please amend your detailed online Financial Disclosure statement. This is published with the article. It must therefore be completed in full sentences and contain the exact wording you wish to be published.

a) State the initials, alongside each funding source, of each author to receive each grant. For example: "This work was supported by the National Institutes of Health (####### to AM; ###### to CJ) and the National Science Foundation (###### to AM)."

c) State what role the funders took in the study. If the funders had no role in your study, please state: “The funders had no role in study design, data collection and analysis, decision to publish, or preparation of the manuscript.”

4. Please update your online Competing Interests statement. If you have no competing interests to declare, please state: “The authors have declared that no competing interests exist.”

5. In the online submission form, you indicated that "data used in this study is available on reasonable request". All PLOS journals now require all data underlying the findings described in their manuscript to be freely available to other researchers, either 1. In a public repository, 2. Within the manuscript itself, or 3. Uploaded as supplementary information.

6. We have noticed that you have uploaded Supporting Information files, but you have not included a list of legends. Please add a full list of legends for your Supporting Information files after the references list.

Additional Editor Comments (if provided):

Reviewers' comments:

Reviewer's Responses to Questions

**Comments to the Author**

1. Does this manuscript meet PLOS Global Public Health’s publication criteria? Is the manuscript technically sound, and do the data support the conclusions? The manuscript must describe methodologically and ethically rigorous research with conclusions that are appropriately drawn based on the data presented.

Reviewer #1: Yes

Reviewer #2: Yes

2. Has the statistical analysis been performed appropriately and rigorously?

Reviewer #1: Yes

Reviewer #2: Yes

3. Have the authors made all data underlying the findings in their manuscript fully available (please refer to the Data Availability Statement at the start of the manuscript PDF file)?

Reviewer #1: Yes

Reviewer #2: Yes

4. Is the manuscript presented in an intelligible fashion and written in standard English?

Reviewer #1: Yes

Reviewer #2: Yes

5. Review Comments to the Author

Reviewer #1: This is an important research study exploring the translation and validation of tools in non-English speaking populations especially those in East Africa which also have a different cultural context to high-income settings where most tools are borrowed from.

The researchers documented the whole process of translation, giving information on the different stages and why they did so. The validation process is also elaborated well.

Overall, this was a well-conducted study, however, a few points are left to clarity for publication.

Comment 1:

Please provide a justification for the sample size you chose of 90 participants.

Comment 2

Table 1 is very confusing. The column labeling is not clear and uniform.

Why is there an empty column between EMPATH and FACES and what do these column labels even mean?

Comment 3

Line 164-165; was additional consent sought for this particular study and not only the main study? State this clearly.

Comment 4

It is not clear which CDQ you are referring to at the different stages it was administered i.e. (the locally adapted one or the original CDQ)

For example, in the criterion validation, the first administration by RA is assumed to be the final version of the CDQ translated to the local language (presumably), and when the physiatrist sees the same patient and administers the DSMV, and fills the CDQ summary sheet. It is not apparent whether it is the translated one or the original one in English. Please clarify.

Comment 5

In assessing the validity of the translated CDQ we compare it with diagnosis by a specialist who is the gold standard in this case. Can you further explore how much the difference in the cadre of the interviewer affects this outcome?

Comment 6

Line 196, which 11 items?? You did not mention them before

Comment 7

Line 244, is it 3 or 4 languages?? You spoke of 4 before, why mention 3

Comment 8

In your discussion, you have not compared the sensitivity, specificity and overall accuracy of the locally adapted CDQ to that of the original English version and explore what it means. Comparison was with a South African study which would do good to come after this.

Reviewer #2: I support publication of this paper if the authors are willing to address the following concerns

1. There is not enough interrogation of the process of forward and back translation of tools developed for the needs of clinicians and patients in non-Western countries. Linguistic factors are important and the authors do consider these carefully. However, culture is more than simply language and language itself is used fluidly. It would be better to situate the paper in a cultural context as well as a linguistic one. I recommend reading the following open source paper Weekes BSH. Aphasia in Alzheimer’s Disease and Other Dementias (ADOD): Evidence From Chinese. American Journal of Alzheimer’s Disease & Other Dementias®. 2020;35. doi:10.1177/1533317520949708

2. There are minor errors of punctuation and spelling. I would be happy to proof read another copy to help.

Brendan Weekes

6. PLOS authors have the option to publish the peer review history of their article (what does this mean?). If published, this will include your full peer review and any attached files.

**Do you want your identity to be public for this peer review?** For information about this choice, including consent withdrawal, please see our Privacy Policy.

Reviewer #1: **Yes: **Dr. Faith Aikaeli

Reviewer #2: No

---

## [Decision Letter · Decision Letter 1]

20 Feb 2024

Adaptation of the Client Diagnostic Questionnaire for East Africa

PGPH-D-23-00103R1

Dear Dr. Kwobah,

We are pleased to inform you that your manuscript 'Adaptation of the Client Diagnostic Questionnaire for East Africa' has been provisionally accepted for publication in PLOS Global Public Health.

Best regards,

Abhijit Nadkarni

Academic Editor

Reviewer Comments (if any, and for reference):

Reviewer's Responses to Questions

**Comments to the Author**

1. If the authors have adequately addressed your comments raised in a previous round of review and you feel that this manuscript is now acceptable for publication, you may indicate that here to bypass the “Comments to the Author” section, enter your conflict of interest statement in the “Confidential to Editor” section, and submit your "Accept" recommendation.

Reviewer #2: All comments have been addressed

2. Does this manuscript meet PLOS Global Public Health’s publication criteria? Is the manuscript technically sound, and do the data support the conclusions? The manuscript must describe methodologically and ethically rigorous research with conclusions that are appropriately drawn based on the data presented.

Reviewer #2: Yes

3. Has the statistical analysis been performed appropriately and rigorously?

Reviewer #2: Yes

4. Have the authors made all data underlying the findings in their manuscript fully available (please refer to the Data Availability Statement at the start of the manuscript PDF file)?

Reviewer #2: Yes

5. Is the manuscript presented in an intelligible fashion and written in standard English?

Reviewer #2: Yes

6. Review Comments to the Author

Reviewer #2: (No Response)

7. PLOS authors have the option to publish the peer review history of their article (what does this mean?). If published, this will include your full peer review and any attached files.

**Do you want your identity to be public for this peer review?** For information about this choice, including consent withdrawal, please see our Privacy Policy.

Reviewer #2: **Yes: **Brendan Stuart Weekes
